# Genome-Wide Characterization and Expression Profiling of the AP2/ERF Gene Family in *Fragaria vesca* L.

**DOI:** 10.3390/ijms25147614

**Published:** 2024-07-11

**Authors:** Yangfan Wei, Yihan Kong, Huiwen Li, Anqi Yao, Jiaxin Han, Wenhao Zhang, Xingguo Li, Wenhui Li, Deguo Han

**Affiliations:** Key Laboratory of Biology and Genetic Improvement of Horticultural Crops (Northeast Region), Ministry of Agriculture and Rural Affairs, National-Local Joint Engineering Research Center for Development and Utilization of Small Fruits in Cold Regions, College of Horticulture and Landscape Architecture, Northeast Agricultural University, Harbin 150030, China; wyfan0614@163.com (Y.W.); kyihan0731@126.com (Y.K.); lhw989874@163.com (H.L.); a02200199@neau.edu.cn (A.Y.); a02140301@163.com (J.H.); 18346971459@163.com (W.Z.); xingguoli@neau.edu.cn (X.L.)

**Keywords:** *Fragaria vesca* L., AP2/ERF, cold, abiotic stresses, transcription factor

## Abstract

The wild strawberry (*Fragaria vesca* L.; *F. vesca*) represents a resilient and extensively studied model organism. While the AP2/ERF gene family plays a pivotal role in plant development, its exploration within *F. vesca* remains limited. In this study, we characterized the AP2/ERF gene family in wild strawberries using the recently released genomic data (*F. vesca* V6.0). We conducted an analysis of the gene family expansion pattern, we examined gene expression in stem segments and leaves under cold conditions, and we explored its functional attributes. Our investigation revealed that the FvAP2/ERF family comprises 86 genes distributed among four subfamilies: AP2 (17), RAV (6), ERF (62), and Soloist (1). Tandem and segmental duplications significantly contributed to the growth of this gene family. Furthermore, predictive analysis identified several cis-acting elements in the promoter region associated with meristematic tissue expression, hormone regulation, and resistance modulation. Transcriptomic analysis under cold stress unveiled diverse responses among multiple *FvAP2*/*ERFs* in stem segments and leaves. Real-time fluorescence quantitative reverse transcription PCR (RT-qPCR) results confirmed elevated expression levels of select genes following the cold treatment. Additionally, overexpression of *FvERF23* in *Arabidopsis* enhanced cold tolerance, resulting in significantly increased fresh weight and root length compared to the wild-type control. These findings lay the foundation for further exploration into the functional roles of *FvAP2/ERF* genes.

## 1. Introduction

The detrimental effects of low temperatures on plant physiology and productivity are significant concerns in agricultural production and ecosystems. According to scientific research, low temperatures can lead to many types of injuries in plants, including freeze, cold, and frost damage [1,2]. These injuries negatively affect plant growth and development and may even lead to plant death [3]. In addition to causing direct damage to plants, low temperatures also affect the metabolic processes and reproductive aspects of plants. Furthermore, low temperatures impair the physiological processes involved in plant resilience, reducing their capacity to withstand and adapt to additional environmental stresses.

There are many families of transcription factors that play critical roles in plant growth, development, and environmental adaptation. One of them is the AP2/ERF family of transcription factors, widely involved in various aspects of the plant life cycle, including, but not limited to, flower, fruit, and seed development, as well as the response to biotic and abiotic stresses [4,5]. Based on the structural and functional characteristics of the genes, this family is meticulously subdivided into several subfamilies, such as AP2, ERF, RAV, and Soloist [6]. Members of each subfamily assume specific biological functions, acting through the regulation of downstream target genes to control plant growth, development, and environmental adaptation in a fine and complex network [7]. Two AP2 structural domains are present in the AP2 subfamily of transcription factors, which are involved in flower development [8], leaf morphology [9], identification of leaf epidermal cells [10], identification of the spikelet meristem [11], embryo development, and seed growth [12]. Members of the RAV subfamily contain one AP2 domain and one B3 domain [13,14]. The RAV subfamily genes mainly respond to the stimulation of ethylene and oleuropein lactone. The *RAV1* and *RAV2* genes in *Arabidopsis* are closely related to leaf development [15,16]. On the other hand, members of the ERF subfamily contain an AP2 structural domain, with ERF subfamily genes playing important roles in plant responses to biotic and abiotic stresses [7]. For example, the *ERF1* and *ERF2* genes in *Arabidopsis thaliana* play important roles in drought and cold stress response. By changing the structure of the rice root system, *OsERF71* improves drought resistance by directly controlling the transcription of genes involved in the synthesis of lignin and cell walls [17]. In addition, *ERF*-like transcription factors are also involved in hormone signaling, e.g., soybean *GmERF3* can be induced by abscisic acid, salicylic acid, jasmonic acid, and ethylene glycol [18]. In the study of Solanaceae, the *SlERFJ2* gene in tomato has been identified as being involved in chlorophyll growth and development using a correspondence database screen [19]. There are 153 ERF subfamily members in eggplants [20], while, in chili peppers, 144 ERF genes have been found, eight of which have been shown to be significantly expressed under treatments with the phytohormones SA, MeJA, and ETH [21]. Additionally, 155 ERF members have been identified in potatoes, with five upregulated in expression under ethylene treatments [22]. In addition to the subfamily members of AP2, ERF, and DREB, there are also some AP2/ERF gene family members that can be associated with cold stress [23,24]. Some AP2/ERF gene family members can bind to cold-stress-related cis elements to regulate the expression of related genes. These cis elements include C-repeat/DRE, LTRE (Low-Temperature-Responsive Element), which can be recognized and bound by members of the AP2/ERF gene family at low temperatures, thereby regulating the expression of related genes.

*F. vesca* (2n = 2x = 14) is a plant of great research value [25,26]. Its genome is relatively small and there is considerable sequence identity between diploid *F. vesca* and octoploid cultivated strawberry [27,28]. In recent years, significant progress has been made in the study of its whole genome. The publication of the latest genomic data of *F. vesca* provides a more accurate database for genetic studies and molecular breeding of strawberry and other Rosaceae plants. We utilized wild strawberry genomic data (*F. vesca* V6.0) (Appendix A) of higher genomic quality [29]. The AP2/ERF gene family was investigated in terms of phylogenetic analysis, conserved motifs, domain, gene structure, chromosomal localization, and covariance analysis, as well as with respect to the gene expression pattern under a low-temperature treatment. One gene family member, *FvERF23*, was identified and shown to be better adapted to low temperatures in plants overexpressing this gene. The goal of this study on the very significant AP2/ERF gene family is to provide the groundwork for future functional research on the AP2/ERF gene family in *F. vesca* and its use in breeding.

## 2. Results

### 2.1. Wild Strawberry AP2/ERF Transcription Factor Family and Phylogenetic Analysis

The latest genomic data of *F. vesca* V6.0 and *Arabidopsis* were used for the Hidden Markov Model (HMM) to search for AP2 structural domains (PF00847), and 220 *Arabidopsis* AP2/ERF genes and 86 *F. vesca* AP2/ERF genes were finally obtained by structural domain screening. Following Nakano et al. [6], by using a classification method based on the AP2/ERF transcription factor AP2 domain sequence of *Arabidopsis thaliana*, this study divided the AP2 superfamily genes of *Arabidopsis thaliana* and *F. vesca* into four subfamilies, each containing 86 AP2 subfamily genes, 206 ERF subfamily genes, 13 RAV subfamily genes, and one Soloist to create an evolutionary tree, with the ERF subfamily further divided into 12 subgroups (ERF I–XII) (Figure 1). Among them, there were 17 AP2 subfamily genes in forest strawberry which accounted for 19.77% of all genes. Six RAV subfamily genes accounted for 6.97% of the total number of genes; we also found 62 ERF subfamily genes and one Soloist gene.

### 2.2. Analysis of the Structural Composition and Physicochemical Properties of the FvERF Genes

The majority of members of the *F. vesca* AP2/ERF gene family (105) exhibited all three motifs simultaneously, according to an analysis of the conserved motifs of the AP2/ERF transcription factors. This suggests that motifs 1, 2, and 5 are the main motifs that make up the structural domain of AP2 (Appendix A). In the meantime, the *F. vesca*’s conserved motif composition of the same subfamily of genes demonstrated a high degree of resemblance and performed comparable biological roles. We suggest that motif 9 and motif 10 are unique to the forest strawberry RAV subfamily, whereas motif 3 is found almost exclusively in the FvAP2 subfamily. These two distinct motifs are thought to carry out more specialized biological tasks. Sequence prediction of AP2/ERF family proteins by NCBI CD search tool showed that all protein sequences contained AP2 structural domains and six proteins contained B3 structural domains and belonged to the RAV subfamily. Seventeen proteins contained two AP2 structural domains and belonged to the AP2 subfamily in terms of classification, while the rest of the proteins belonged to the ERF subfamily (Appendix A). The structural composition of the genes showed that 51.16% of the genes contained only one exon and the rest of the genes contained multiple exons. More interestingly, four RAV members did not contain introns. Additionally, the exons in all 17 AP2 members were separated by multiple introns (Appendix A). It appears that genes located in the same subfamily also have extremely similar structural features, so there may be consistency in the exercise of protein function [30]. Physicochemical property prediction showed that the sequence of FvERF-encoded proteins was 134–826 aa, and the proteins exhibited molecular weights between 15.1 kDa and 89.8 kDa. The majority of the FvERF proteins included basic amino acids, while their isoelectric point (PI) distribution ranged from 4.63 to 9.81. The proteins of FvERFs were all categorized as unstable proteins, since their instability coefficients ranged from 28.7 to 72.7. The aliphatic amino acid indices were distributed between 32.57 and 79.84, and their thermal stability varied widely. The total hydrophilicity of the proteins was less than 0, and all were hydrophilic. The subcellular localization of the FvERFs proteins was also predicted and it was found that 71 proteins were located in the nucleus, whereas 9 proteins were located in the chloroplasts, FvERF55 was located in the cytoskeleton, FvERF22 was located in the mitochondria, and FvERF45 was located in the peroxisomes. The structural characterization, physicochemical features, and subcellular localization predictions of individual members of the FvERF gene family are detailed in the Appendix A.

### 2.3. Chromosome Localization and Gene Family Evolutionary Analysis

The members of the *FvERF* gene family were given new names based on their chromosomal order (Appendix A). The localization of strawberry *FvERF* gene family members on chromosomes was analyzed, and, as shown in Figure 2, the distribution of the 86 genes on chromosomes chr1–7 exhibited a heterogeneous trend. For instance, with 18 genes on the chr6 chromosome, the latter exhibited the highest density of *FvERF* genes. The number of *FvERF* genes on chr1 and chr3 was similar, with 8 and 7, respectively (Figure 2). The distribution of the remaining genes on the other chromosomes was also heterogeneous and did not correlate with chromosome length.

Genes are replicated in smaller regions through two mechanisms: tandem duplication and reverse transcription transposition [31]. We used Circos analysis to better understand the *FvERF* gene families expansion pattern, and the results revealed that 36 genes were distributed throughout the chromosome, 10 genes belonged to tandem duplications, and 31 genes belonged to WGD or segmental [32], suggesting that tandem duplications and segmental duplications occupy important roles in addition to genome-wide duplications of these genes, causing the expansion of the *FvERFs* gene family (Figure 3). *F. vesca* is the ancestral diploid of the octoploid strawberry *F.* × *ananassa* [33], and we investigated covariation among the model crop *Arabidopsis thaliana*, the diploid *F. vesca*, and the octoploid strawberry *F.* × *ananassa* species. The One Step MCscanX software (https://github.com/CJ-Chen/TBtools/ (accessed on 7 July 2024)) program identified 77,126 repeat gene pairs between *F. vesca* and *F.* × *ananassa*, and 12,117 repeat gene pairs between *Arabidopsis thaliana* and *F. vesca*. Among the three species, 48 members of the AP2/ERF gene family exhibited collinearity (Figure 4; Appendix A).

### 2.4. AP2/ERF Promoter Region Cis-Acting Element Exploration

PlantCARE was utilized to anticipate cis-acting elements in a 2 kb sequence that was retrieved and uploaded upstream of the *FvAP2/ERF* gene member start codon (ATG). When the cis-acting elements in the promoters of the *FvAP2/ERF* genes in *F. vesca* were analyzed, it was found that the majority of the cis-acting elements were light-responsive elements. Almost every promoter region of the genes contained at least three light-responsive elements and were not, therefore, considered. There is a large proportion of hormone-responsive elements, including cis-acting elements in response to ABA, auxin, and MeJA, among others. There are also many cis-acting regulatory elements involved in phloem expression, seed specificity, and cell-cycle regulation. There are also many cis-acting regulatory elements involved in meristem expression, seed specificity, and cell cycle regulation. It is noteworthy that some MYB binding sites were found to be involved in drought-induced and low-temperature-responsive cis-acting elements (Appendix A).

### 2.5. FvERF Genes Expression Profile in the Cryotranscriptome

In response to the previous section, we chose to further explore the transcriptomic data of the *FvERF* gene under cold conditions. *F. vesca* plants were subjected to a freezing treatment (−4 °C) and a chilling treatment (4 °C) for 6 h. Stem and leaf tissues from those strawberry plants were compared with those from the room temperature (22 °C) control group. The RNA sequencing (RNA-Seq) results of the AP2/ERF genes were screened by transcriptome sequencing, and the data were analyzed by clustering methods. The clustering results are shown in Figure 5. Compared with the control, 22 *FvERFs* showed significant expression after the cold treatment. However, the expression levels of other genes were either minimal or not significantly different from those exhibited by the control. Additionally, the results within different clustering subgroups varied. In order to verify the reliability of the RNA-Seq results, we cold-treated *F. vesca* plants, collected their stem and leaf tissues, and designed specific primers for 10 of the 22 significantly expressed genes to check their expressions by RT-qPCR. The expression of these ten genes is shown in Figure 6, and the expression levels of the genes generally increased after the cold treatment (freezing and chilling) compared with the control, further validating the response of the wild strawberry *FvERFs* gene to cold temperatures.

### 2.6. Expression Patterns of Cold-Stress-Responsive Genes in Strawberry

Many genes are involved in the regulation of cold tolerance signaling in plants. In addition to the classic ICE–CBF–COR cold signaling pathway, ERF family genes can also bind to the promoter of the *CBF* gene to activate downstream expression of *COR* genes and improve cold tolerance in plants [34]. In this study, we cloned four cold-responsive genes (*FvCBF1*, *FvCBF3*, *FvCOR15a*, *FvRD29A*) from *F. vesca* and used them to study the gene expression patterns under low temperatures. The expression of the first two genes (*FvCBF1*, *FvCBF3*) increased rapidly after 0.5 h of cold treatment and increased with the prolongation of the cold treatment duration, and the overall level was maintained at a high level (Figure 7A,B). The expression of the last two genes (*FvCOR15a, FvRD29A*) decreased after 0.5 h of cold treatment compared with the control, before starting to increase slowly with the prolongation of the cold treatment duration and reaching the extreme point after 24 h and 12 h, respectively (Figure 7C,D). These results suggest that *FvCBF1*, *FvCBF3*, *FvCOR15a*, and *FvRD29A* are involved in cold stress response.

### 2.7. Overexpression of FvERF23 in Arabidopsis thaliana

In this study, we found that *FvERF23* and *AtERF66* belong to the same evolutionary clade (Figure 1) and that 48 genes share interspecies collinearity among three species (including *FvERF23*) (Appendix A). Additionally, compared with the control roots and stems, the expression levels of stems and leaves of the plants were significantly increased after the cold treatment. This finding is consistent with the transcriptome data. After 6 h of cold treatment, *FvERF23* was significantly activated and its expression level was the highest, indicating that *FvERF23* responded most strongly to cold stress. Therefore, we selected *FvERF23* as a candidate gene to further explore its function. We obtained five *Arabidopsis* plants overexpressing *FvERF23* by using the *Agrobacterium*-mediated floral dip method, we extracted their DNA for PCR verification, and we selected three of them (L1, L4, and L5) with the highest expression to be subjected to the cold treatment (Appendix A). It was found that, after 7 days of growth of *Arabidopsis* seeds, root length and fresh weight were generally reduced in low-temperature-treated plants compared with the control, but the average root length and fresh weight of the low-temperature-treated transgenic plants were greater than those of WT (Figure 8A–C). It is noteworthy that, after 28 d of plant growth, the proline and MDA contents of all the cold-treated lines increased and the proline content of the transgenic *Arabidopsis* was greater than that of WT, although not as large as the MDA content of WT (Figure 8D,E).

The phenotypes of wild-type *Arabidopsis thaliana* and transgenic *Arabidopsis thaliana* were observed, and antioxidant enzyme activities were measured under different stresses. There was no difference between the two in the natural growth state, but the leaves of the wild-type *Arabidopsis* were more water-stained after the cold treatment, whereas only a few leaves of the transgenic *Arabidopsis* showed water-staining and were less damaged. Overexpression of *FvERF23* in *Arabidopsis thaliana* increased tolerance to low temperatures, salt, drought, and ABA (Figure 9A). Antioxidant enzyme activities are closely related to plant stress tolerance, and, together, they are involved in regulating plant stress response, growth, and development to ensure plant survival and growth in harsh environments. The enzyme activities of POD and SOD were measured in normal and treated *Arabidopsis thaliana*, and it was found that the enzyme activity of the treated plants experienced an increase, but the POD enzyme activity of the transgenic *Arabidopsis thaliana* was higher than that of the wild type, with the largest difference from the wild type in the cold treatment, followed by the ABA treatment; the SOD enzyme activity showed similar changes (Figure 9B,C). The results show that the stress tolerance of plants characterized by an overexpression of *FvERF23* was significantly improved, and *FvERF23* might also be involved in the response to salt, drought, and ABA.

## 3. Discussion

As one of the biggest plant gene-specific families, AP2/ERF is important for both gene regulation and response to external stimuli [35]. It also plays a role in plant growth and development. *F. vesca* has great research value as a model diploid species of strawberries. The publication of the complete genome of the model species, strawberry, has greatly facilitated the progress of genetics research and molecular breeding in strawberries. With the application of novel sequencing technologies, relevant researchers have reassembled and annotated the *F. vesca* genome. They have obtained the latest complete genome of *F. vesca* v6.0. Based on this study, the AP2/ERF transcription factor family from *F. vesca* was analyzed and characterized, and a total of 86 members were identified, including 17 subfamily AP2 members, 62 ERF subfamily members, 6 RAV subfamily members, and one Soloists subfamily member. Among other plant genomes that have been sequenced, *Arabidopsis* (125 Mb) contains 147 AP2/ERF genes [36], maize (2.3 Gb) contains 292 AP2/ERF genes [37], and grape (475 Mb) contains 132 AP2/ERF genes [9]. This shows that there is no consistency between genome size and AP2/ERF gene membership across species.

Diversity in protein motifs and gene structures underlies the evolution of gene families. Motif analysis of *F. vesca* AP2/ERF gene families revealed similar motif composition among the same subfamilies. Motifs 1 and 2 were present in all members of the family, while only motifs 3 and 10 were found in the AP2 and RAV subfamilies, respectively. These molecular features give the *FvAP2* and *FvRAV* genes unique roles that set them apart from the other genes [38]. The AP2/ERF family of transcription factors is characterized by the presence of a single AP2 structural domain, each containing 60–70 amino acid residues, and the typical helical structure is responsible for the specific binding of sequence nucleic acids to target genes to regulate expression [39]. It contains two elements named YRG and RAYD [40]. The YRG element consists of approximately 20 amino acid residues and is an alkaline hydrophilic region consisting of three β-folds, where the amino acids present at positions A14 and D19 are located on the second β-fold, which plays a crucial role in the specific binding of the transcription factor to the cis element [41]. The RAYD element consists of approximately 40 amino acid residues and is an amphipathic region. This region has an α-helix and the α-helix consists of 18 amino acids. RAYD elements may be key elements in the interaction and cooperation among transcription factors [36]. The intron number is associated with gene transcription efficiency and evolution at the gene level [38]. The greater the number of introns, the greater the chance of generating selective shearing. The results from the study of the higher number of introns contained in genes within the AP2 subfamily also appeared to be consistent across other species, such as *Perilla frutescens* [42], *Gynostemma pentaphyllum* [43], radish [44], and quinoa [45].

The chromosomal localization map results demonstrated that the *FvAP2/ERF* gene density was not distributed uniformly across the seven chromosomes, a phenomenon which was associated with genome-wide duplication of the genes. The number of *F. vesca FvAP2/ERF* genes on chr 1 and chr 3 was almost the same, but their lengths differed considerably. Some of the genes on chr 2, chr 4, and chr 7 formed gene clusters, which are associated with tandem duplication of genes [46]. Gene duplication affects chromosome location, while gene amplification depends on sequence duplication [47]. The occurrence of tandem and segmental duplication events within this study suggests that tandem and segmental duplications have a greater impact on gene evolution in *F. vesca* AP2/ERF genes, in addition to gene diversity generated by genome-wide duplications which have become the main drivers of gene evolution. The interspecies collinearity analysis found 12,117 and 77,126 immediate homologous gene pairs between *F. vesca* and *Arabidopsis* and *F. vesca* and *F.* × *ananassa*, respectively, indicating that several *FvAP2/ERF* gene pairs were generated from a single gene through intergenomic duplication. Notably, only 48 members of the AP2/ERF gene family exhibited interspecies collinearity with all three species simultaneously, a phenomenon which also lays the foundation for investigating gene functions among different species.

Promoter analysis is an important method for identifying regulatory networks between environmental stimuli and gene expression [48]. In order to reflect the potential activities of the genes and find potential regulatory responses to abiotic stimuli and hormones, members of the *FvAP2/ERF* gene family were searched for cis elements in this study. Wan et al. [49] identified an ERF subfamily protein, *OsDERF1* (drought-responsive ERF genes), which can directly bind to the GCC-box or DRE elements in the promoters of *OsERF3* and *OsAP2–39,* activate their gene expression, and negatively regulate drought tolerance in rice. Our focus was on stress and hormone-related response elements among the 86 genes exhibiting multiple response elements. Numerous hormone-associated elements that respond to ABA, auxin, MeJA, and other hormones were found in the promoter region of the *FvAP2/ERF* genes. These elements may be crucial for plant growth. The great majority of *FvAP2/ERF* genes have several cis-acting elements, including anaerobic stress elements (ARE), drought stress response elements (DRE), and cold response elements (LTR), and they are all strongly linked to the plant’s stress response. It is hypothesized that *F. vesca* AP2/ERF genes play important roles in abiotic stress processes. While cis-acting elements varied among different family members, similarities suggest functional differentiation and synergistic regulation of AP2/ERF transcription factors within the family [50].

Cold RNA-seq data of *F. vesca* stems and leaves showed significant expression changes in certain genes after the cold treatment compared to the control. Gene expression patterns provide clues as to how a gene functions. It has been shown that overexpression of *VvERF63* improves cold tolerance in grape leaves; however, cold tolerance declines after gene silencing [51]. The 10 genes obtained after screening were used to analyze expression patterns after exposure to cold stress. This study shows that the *FvERF23*, *FvERF2*, *FvERF25*, and *FvERF23* genes in leaves, as well as the *FvERF23*, *FvERF41*, *FvERF25*, and *FvERF23* genes in stem segments, strongly responded to low temperatures. *BpERF13* can bind to the *CBF* gene promoter and, thus, regulate the expression of downstream *COR* genes. This regulatory mechanism can activate the plant cold response and improve the cold tolerance of plants [34]. In this experiment, four cold-responsive genes, *FvCBF1*, *FvCBF3*, *FvCOR15a*, and *FvRD29A*, were cloned from strawberry plants and it was found that the expression levels of the first two genes increased rapidly after the cold treatment and reached their peaks after 12 h, while the last two genes reached their peaks later and exhibited lower expression levels. The results suggest that *ERF* genes are involved in transcriptional activation and regulation of downstream genes and play an important role in this process. Further research into the regulation mechanism of genes under cold conditions is necessary, as multiple other genes exhibited varying degrees of responsiveness. In this experiment, following the overexpression of *FvERF23* in *Arabidopsis thaliana*, the plants did not exhibit significant differences under room-temperature control conditions, but their root length was significantly suppressed compared to that found in the cold treatment group. Its overexpression in *Arabidopsis* resulted in a significantly greater fresh weight and root length compared to the wild type. It has been observed that low temperatures break plant dormancy, leading to a decrease in the expression level of *PpcAP2/ERF* genes, which regulate cherry bud growth. This decrease occurs due to a reduction in the abundance of many *PpcERF* gene transcripts during plant dormancy which promotes, to a certain extent, the lifting of endodormancy and, thereby, influences the plant growth status [47].

Proline is an important amino acid widely found in plants which accumulates in the cytoplasm, cell wall, and vesicles and regulates the osmoregulatory pressure within the cell; it also acts as an antioxidant, scavenging reactive oxygen radicals in the cell and reducing damage from oxidative stress [52]. MDA, on the other hand, is a product of oxidative stress and is the result of the reaction of lipid peroxidation in the cell membrane [53]. The phenotypic differences between overexpressed *FvERF23* plants and wild-type plants after the cold treatment were not significant, but the proline content of the former was higher than that found in the wild-type plants and the MDA content of the overexpressed *FvERF23* plants was lower than that found in the wild-type plants. These results indicate that the transgenic plants had improved cold tolerance. We also investigated the resistance of *FvERF23* transgenic *Arabidopsis* to three other stresses. POD and SOD, two important antioxidant enzymes, regulate redox homeostasis by scavenging reactive oxygen radicals and by degrading hydrogen peroxide, promoting the growth and development of plants and enhancing their resistance to adversities. POD and SOD activity was higher than that recorded in the wild type after overexpression of *FvERF23* in response to different stresses. It can be seen that *FvERF23* can induce antioxidant enzyme activity and enhance the resistance of plants to stress. The mechanism of *FvERF23*-regulated gene response to cold stress remains to be explored further.

## 4. Materials and Methods

### 4.1. Determination of the Wild Strawberry AP2/ERF Gene Family

*Fragaria-vesca*-related genome assembly information was downloaded from the Genomic Database for Strawberries (GDS, http://eplant.njau.edu.cn/strawberry (accessed on 7 July 2024)). The HMM of the AP2 structural domain (PF00847) was downloaded from the Pfam Protein Family Database, and the HMM results were extracted using TBtools v2.034 with a cutoff threshold of E-value ≥ 0 [54]. The HMM results were then analyzed by the National Center for Biotechnology Information (NCBI) CD search functions (https://www.ncbi.nlm.nih.gov/Structure/cdd/wrpsb.cgi (accessed on 7 July 2024)) and, with the help of a manual search for the integrity of AP2 structural domains, the redundant genes were eliminated and the target gene family candidate proteins containing at least one AP2 structure were obtained. *Arabidopsis* genomic data and annotation information were obtained from NCBI (https://www.ncbi.nlm.nih.gov/genome (accessed on 7 July 2024)), and the *Arabidopsis* AP2/ERF gene family protein sequences were obtained following the same procedure. The complete amino acid sequences of *F. vesca* AP2/ERF and the screened *Arabidopsis* were stored in the FASTA format, and sequence comparison and phylogenetic tree analysis were performed using MEGA 11 to construct a neighbor-joining (NJ) tree with 1000 bootstrap replicates. The visualization of the evolutionary trees was enhanced using iTOL v6 (https://itol.embl.de/ (accessed on 7 July 2024)), and the AP2/ERF superfamilies (AP2, ERF, RAV TF families) were classified.

### 4.2. Conserved Motifs, Domain, Gene Structure, and Physicochemical Properties

Conserved motifs were identified by the online MEME (http://meme-suite.org/tools/meme (accessed on 7 July 2024)) program by using default parameters and by searching motif numbers of 10. The evolutionary tree, motif, domain, and gene structure were graphically visualized by using TBtools, while the molecular weight, theoretical isoelectric point, instability index, aliphatic index, and total hydrophilicity average were evaluated for FvERF proteins. Subcellular location predictions were analyzed using WoLF PSORT (https://wolfpsort.hgc.jp/ (accessed on 7 July 2024)) online URL.

### 4.3. Chromosomal Localization and Collinearity Analysis of Genes

The genome annotation file of *F. vesca* v6.0 was downloaded from the GDS database, and the chromosomal localization of *FvERFs* genes was visualized using the TBtools software. The genes localized on the chromosome were renamed sequentially. The genome of *F. vesca* was compared with itself, the genome files were processed using the Multiple Covariance Scanning Toolkit (MCScanX), with the parameters set to the default value of E-value ≤ 1 × 10^−10^, the obtained files were subjected to the tandem duplicated genes visualization and segmented duplicated genes visualization, and the Circos plots were constructed. By comparing the genomes of wild strawberries with those of *Arabidopsis* and cultivated strawberries, homologous genes among these three can be screened, and a collinearity analysis among species can be constructed.

### 4.4. Study of Cis-Acting Elements

The target gene family’s 2 kb promoter sequence upstream of the transcription start site was extracted from the *F. vesca* genome annotation file and sequence file using the PlantCARE (http://bioinformatics.psb.ugent.be/webtools/plantcare/html/ (accessed on 7 July 2024)) database, and it was then uploaded to the website for the prediction of cis-action regulatory elements. The obtained data were processed and visualized.

### 4.5. Transcriptome Data Processing and Expression Profiling

Stem and leaf tissues were collected from “Hawaii” strawberry plants grown for 75 days at room temperature (22 °C) and at 4 °C and −4 °C for 6 h. The experiment was repeated three times. The collected samples were subjected to transcriptome sequencing, aiming to mine the cold-resistance-related genes of *FvERFs*. Sequenced data underwent quality assessment with FastQC (version: 0.11.9) and were aligned to the *F. vesca* reference genome (Appendix A) using Hisat2 (version: 2.2.1) to map reads. Gene expression levels of AP2/ERF genes under various treatments were quantified using StringTie’s cufflinks (version: 2.2.1), measuring expression as exon model fragments per kilobase per million mapped reads (FPKM). The expression data of the corresponding genes were obtained by comparing the genomic versions. FPKM + 1 was logarithmically transformed to log2 and plotted as a heatmap with the help of TBtools [55]. The red-to-blue gradient indicates the expression level from high to low.

### 4.6. Plant Material and Real-Time Fluorescence Quantitative PCR Validation

Primers were designed according to the *FvERF23* coding sequence (CDS) region and the gene was amplified using PCR. Subsequently, a recombinant plasmid was constructed by linkage to the pCAMBIA1300-GFP vector, and the transgenic plants were obtained by *Agrobacterium* (GV3101)-mediated transformation of the Columbia ecotype (Col) *Arabidopsis thaliana* [56]. The transgenic *Arabidopsis* plants were screened by selecting 1/2 Murashige and Skoog (MS) medium + 50 mg/L kanamycin, and, finally, DNA was extracted for PCR verification [57]. The initial root length and fresh weight of the plants were measured after around 7 days of growth. Then, they were transplanted into culture pots containing nutrient-enriched soil and vermiculite, and, after 28 days of growth, the leaves of all plants were collected and stored in liquid nitrogen; the untreated leaves were used as a control to study the physiological changes in the plants before and after the low-temperature treatment. Proline content and malondialdehyde (MDA) content were determined in all the plants. Proline content and MDA content were measured according to the colorimetric method [58].

Total RNA from *Arabidopsis* tissue samples was extracted using the FastPure Cell/Tissue Total RNA Isolation Kit V2 (Vazyme, Nanjing, China), and cDNA was synthesized with the HiScript II One Step RT-qPCR SYBR Green Kit (Vazyme, China). The procedure was carried out according to the product instructions. ChamQ Universal SYBR qPCR Master Mix Kit (Vazyme, China) was used with a 20 µL volume containing 10 µL of mix premix, 2 µL of gene-specific primers (Appendix A), 6 µL of ddH_2_O, and 2 µL of cDNA. RT-qPCR was configured using the subsequent parameters: 40 cycles of 95 °C for 30 s, 95 °C for 5 s, and 60 °C for 30 s (ABI 7500) [59]. The experiments were repeated three times with the *AtActin* gene as a standardized internal control [60], and the data were analyzed using the 2^−ΔΔCt^ method [61].

### 4.7. Plant Growth Stress Experiment

Growth stress experiments were performed on wild-type *Arabidopsis* and *FvERF23*-overexpressing *Arabidopsis* after 40 days of growth. *Arabidopsis thaliana* was treated at −4 °C for 6 h to simulate the low-temperature stress during growth. The plants were irrigated with 300 mM NaCl for 5 d to simulate salt stress, and 10% PEG-6000 and 250 µM ABA solutions were used to simulate drought stress and ABA stress, respectively. The phenotypes of the plants were observed before and after the treatments. The leaves of *Arabidopsis thaliana* were collected before and after the stress treatments and peroxidase (POD) and superoxide dismutase (SOD) activity was measured. POD and SOD activity was determined using the guaiacol method and the nitrogen blue tetrazolium (NBT) photochemical reduction method, respectively [62,63].

## 5. Conclusions

To sum up, we found 86 AP2/ERF genes in *F. vesca*. These genes were divided into four subfamilies, AP2, RAV, ERF, and Soloist, including 17, 6, 62, and 1 genes, respectively, based on the partitioning of conserved structural domains. These 86 genes were dispersed randomly across seven chromosomes. There were similarities in the motif, domain, and gene structure of the same gene subfamily. In addition, chromosome collinearity analysis was conducted to better understand gene expansion and evolution. Cis-acting element predictions provide clues for understanding gene function. Finally, cold transcriptome data were analyzed, revealing that certain genes were highly responsive to cold stress. RT-qPCR was performed to verify and identify a gene that positively responded to cold stress. *FvERF23* significantly improved the cold tolerance of the transgenic plants. The research remains at the stage of verifying gene function, rather than practical application. Therefore, it is necessary to systematically elucidate the functions, mechanisms, and molecular regulatory networks of AP2/ERF transcription factors in response to adversity and apply them to the improvement of strawberry stress tolerance through molecular breeding.

## Figures and Tables

**Figure 1 ijms-25-07614-f001:**
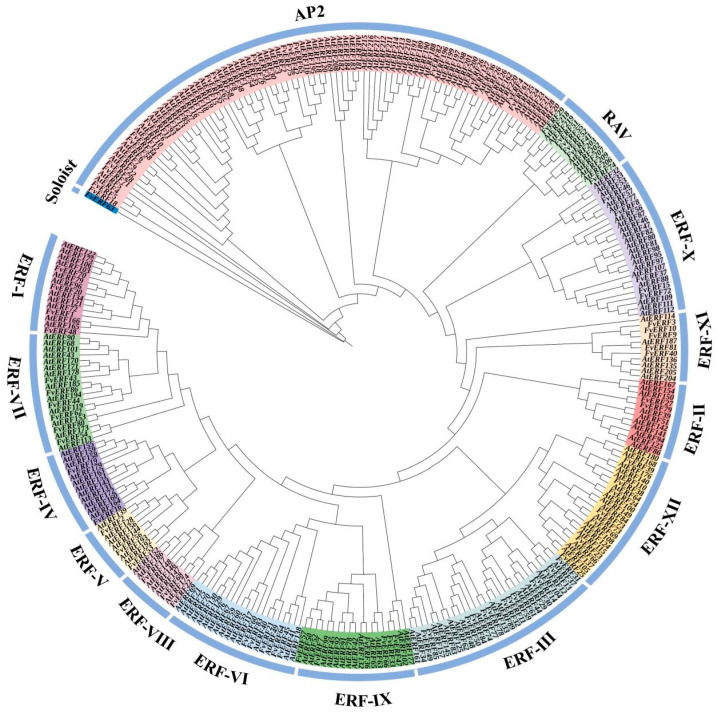
AP2/ERF superfamily phylogenetic evolutionary tree of *Arabidopsis thaliana* and *Fragaria vesca*. The color blocks representing the AP2, ERF (I-XII), RAV, and Soloist subfamilies are depicted uniquely. Names are assigned based on their chromosomal positions. Fv, *Fragaria vesca* L.; At, *Arabidopsis thaliana*.

**Figure 2 ijms-25-07614-f002:**
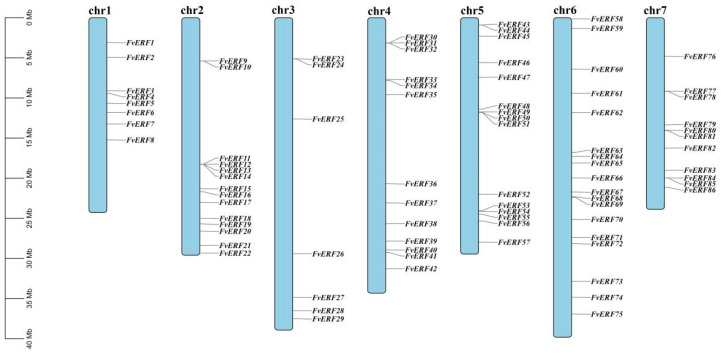
Distribution of AP2/ERF gene locations on chromosomes in *Fragaria vesca*.

**Figure 3 ijms-25-07614-f003:**
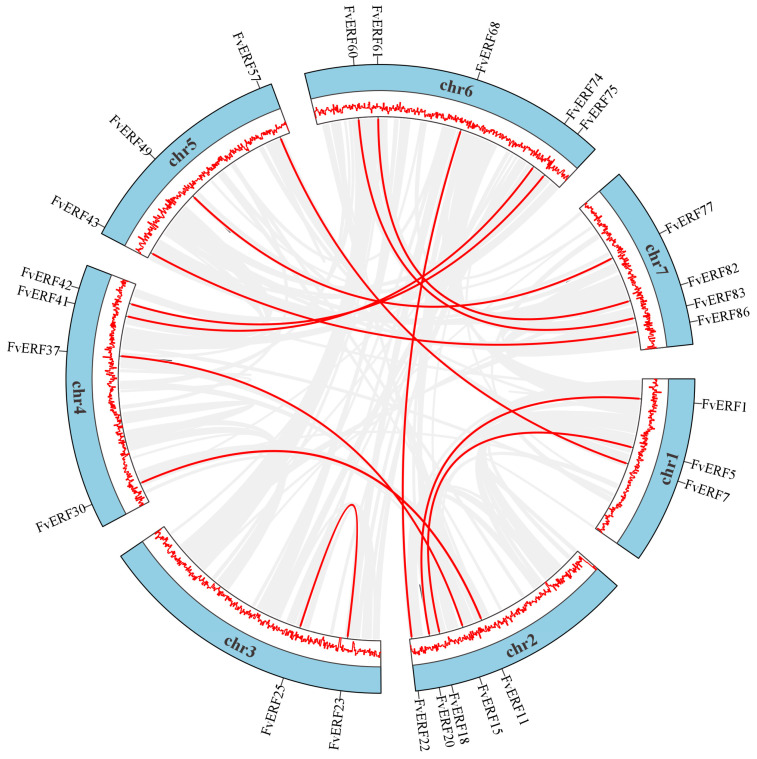
Species collinearity analysis and repeat gene pairs in *Fragaria vesca*. The red line in the center of the figure represents the repeat gene pair, and the gray part represents the collinearity block.

**Figure 4 ijms-25-07614-f004:**
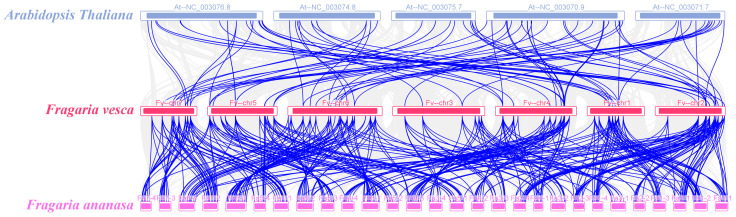
*Arabidopsis thaliana*, *F*. × *ananassa*, and *F. vesca* interspecies collinearity analysis. The bright blue lines represent the collinearity gene pair (see Appendix A), and the gray part represents the collinearity region blocks of the genome of different species.

**Figure 5 ijms-25-07614-f005:**
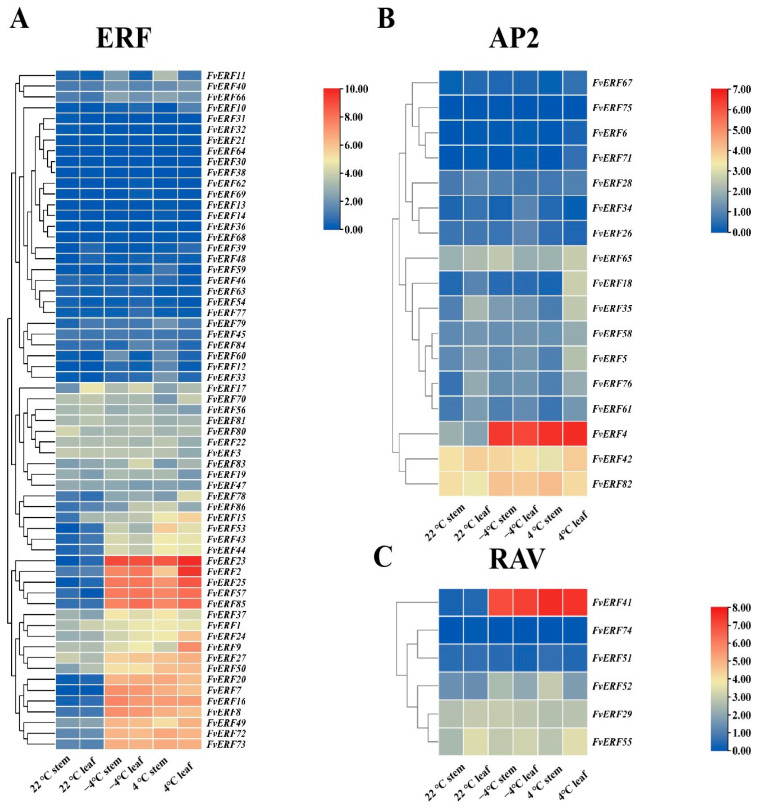
Expression profile of *FvERF* genes in strawberry stem segments and leaves after room temperature (22 °C), freezing (−4 °C), and chilling (4 °C) treatments. Heat map of (**A**) ERF subfamily genes, (**B**) AP2 subfamily genes, and (**C**) RAV subfamily genes. Row data were clustered. The RNA-seq expression data of the corresponding genes, FPKM + 1, were converted with log2 as the base, and the heat map was plotted. Blue to red indicates low to high expression levels.

**Figure 6 ijms-25-07614-f006:**
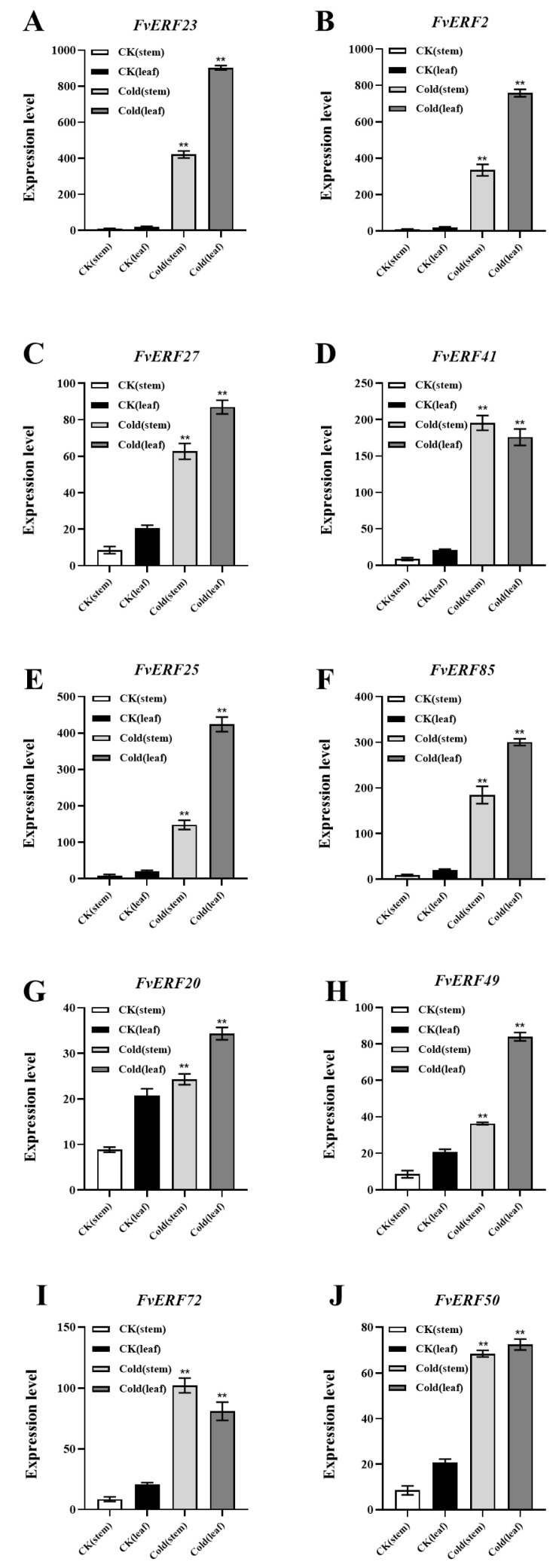
Expression levels of 10 *FvERF* genes (**A**–**J**) from strawberry stem and leaf tissues after different treatments were verified by RT-qPCR. CK: untreated plants; cold: 6 h at 4°C. Every data point was computed utilizing the 2^−ΔΔCt^ technique. The mean of three replicates ± SE is included in the experimental data. A student’s *t*-test was used. *p* ** < 0.01 indicates a highly significant difference.

**Figure 7 ijms-25-07614-f007:**
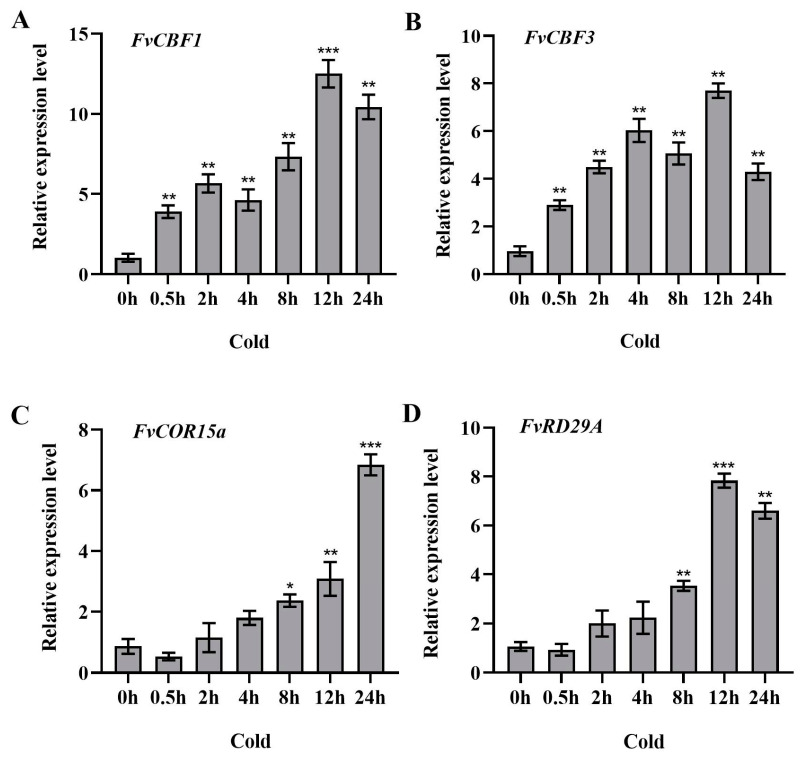
Expression of four genes ((**A**) *FvCBF1*, (**B**) *FvCBF3*, (**C**) *FvCOR15a*, (**D**) *FvRD29A*) in *F. vesca*. was analyzed with the extension of the cold treatment duration. The expression levels of the genes were detected by RT-qPCR and the experiment consisted of three replicates. One-way ANOVA was used. *p* ** < 0.01 and *p* *** < 0.001 indicate a highly significant difference, *p* * < 0.05 indicates a significant difference.

**Figure 8 ijms-25-07614-f008:**
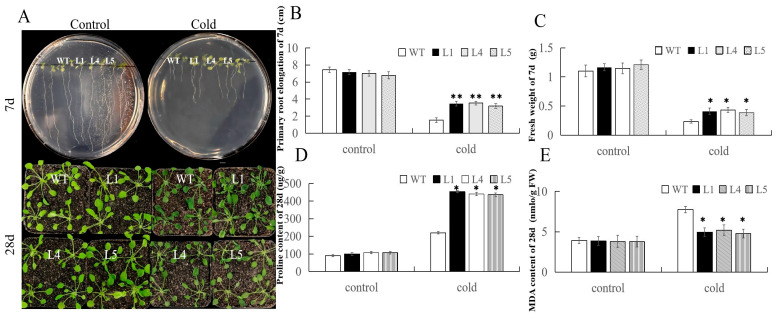
Phenotypes of *Arabidopsis* after 7 days and 28 days of growth under cold stress. (**A**) *Arabidopsis* overexpressing the *FvERF23* gene showed phenotypic differences from WT after cold stress. A student’s *t*-test was used to analyze root length (**B**) and fresh weight (**C**), proline content (**D**), and MDA content (**E**) after 28 days of growth compared with control WT. A student’s *t*-test was used. *p* ** < 0.01 indicates a highly significant difference, *p* * < 0.05 indicates a significant difference.

**Figure 9 ijms-25-07614-f009:**
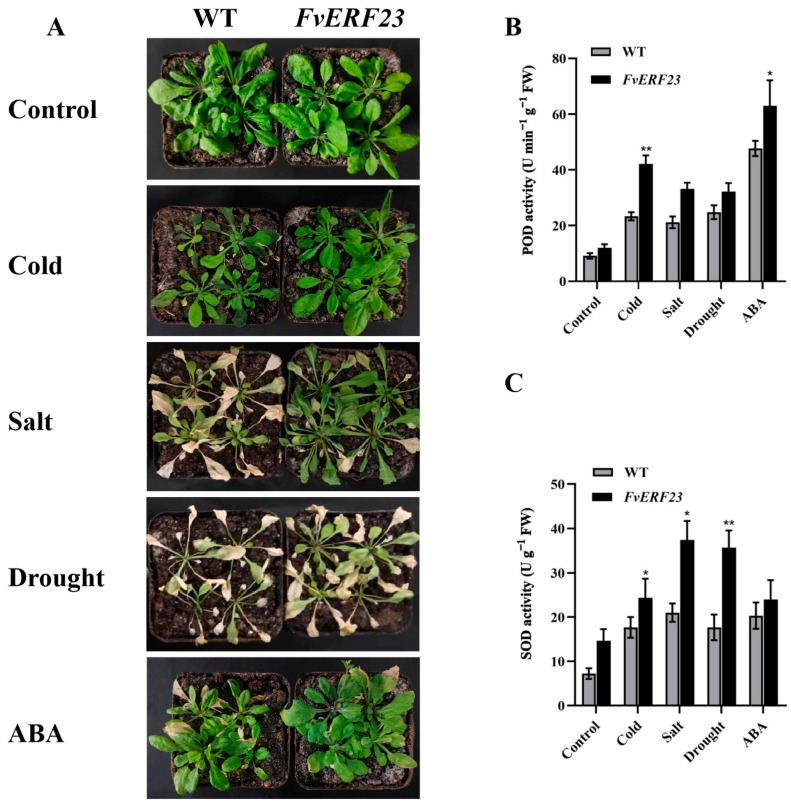
Phenotypic and antioxidant enzyme activity analysis of wild-type and transgenic *Arabidopsis thaliana* under different treatments. (**A**) −4 °C for 6 h to simulate low-temperature stress during its growth. Plants were irrigated with 300 mM NaCl, 10% PEG-6000, and 250 µM ABA solutions for 5 days to simulate salt stress, drought stress, and ABA stress, respectively. (**B**) POD activity and (**C**) and SOD activity were determined for the strains in all treatments. Significance markers were placed by using a one-way ANOVA. *p* ** < 0.01 indicates a highly significant difference, *p* * < 0.05 indicates a significant difference.

## Data Availability

Data are contained within the article or Appendix A.

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
