# Peer review of "Genome-Wide Characterization and Expression Profiling of the AP2/ERF Gene Family in *Fragaria vesca* L."

_ijms, 2024, doi:10.3390/ijms25147614_

Round 1

Reviewer 1 Report

Comments and Suggestions for Authors

The wild strawberry has great research value as a model diploid species of strawberries. AP2/ERF is important for both gene regulation and response to external stress. The study conducted bioinformatics analysis on the AP2/ERF transcription factor family in wild strawberry, and further examined the expression of the FvAP2/ERF family genes under cold conditions. Based on these results, the authors chosen a gene that responds most strongly to cold stress and found that the overexpression of FvERF23 in Arabidopsis enhanced cold tolerance. These findings lay the foundation for future functional research on the AP2/ERF gene family in wild strawberry. However, some parts of the manuscript need further revision.

1. The authors performed overexpression research on FvERF23 in the model plant Arabidopsis but not strawberry, which is very regrettable. Is this because strawberry genetic transformation is very difficult at present?

2. The protein and/or gene expression level of FvERF23 overexpression materials were not detected in this study, and it is uncertain whether it is truly overexpressed. The data needs to be added.

3. As mentioned in Result 1 (Page 2, Line 96-97), “this study divided the AP2 superfamily, genes of Arabidopsis thaliana and F. vesca into four subfamilies, each containing 86 AP2 subfamily genes, 206 ERF subfamily genes, and 13 RAV subfamily genes, respectively”. Here, AP2 superfamily is divided into 4 subfamilies, but only the types and quantities of genes included in the 3 subfamilies are mentioned. Please check it.

4. The authors should better conduct a significant difference analysis on the data in Figure 9. 

5. All references in the manuscript should be superscripted, and there are multiple reference errors in the text, such as on Page 2, Lines 50 and 59.

6. qRT-PCR should be RT-qPCR.

Comments on the Quality of English Language

need ask for a native speaker to polish the language. 

Author Response

Comments 1. The authors performed overexpression research on FvERF23 in the model plant Arabidopsis but not strawberry, which is very regrettable. Is this because strawberry genetic transformation is very difficult at present?

Response 1: Thank you for pointing this out, we agree with you, yes, there are some difficulties in gene transformation in strawberry, but fortunately our lab combined with the previous research currently figured out the stable genetic transformation in strawberry, have overexpression studies on other genes, and initially got the proof at the gene level. We consider to continue to study the transformation and functional analysis of FvERF23 gene in strawberry in our subsequent research. 

Comments 2. The protein and/or gene expression level of FvERF23 overexpression materials were not detected in this study, and it is uncertain whether it is truly overexpressed. The data needs to be added.

Response 2: Thank you very much for pointing this out and we agree with this view. The identification of the expression levels of the FvERF23 gene we provide in the Non-published Material of the corrected manuscript.

Comments 3. As mentioned in Result 1 (Page 2, Line 96-97), “this study divided the AP2 superfamily, genes of Arabidopsis thaliana and F. vesca into four subfamilies, each containing 86 AP2 subfamily genes, 206 ERF subfamily genes, and 13 RAV subfamily genes, respectively”. Here, AP2 superfamily is divided into 4 subfamilies, but only the types and quantities of genes included in the 3 subfamilies are mentioned. Please check it.

Response 3: Thank you for your comment, which we have corrected in line 100 of the revised draft and highlighted in red.

Comments 4. The authors should better conduct a significant difference analysis on the data in Figure 9. 

Response : Thank you for your suggestion and we have modified Figure 9.

Comments 5. All references in the manuscript should be superscripted, and there are multiple reference errors in the text, such as on Page 2, Lines 50 and 59.

Response 5: Thank you for pointing this out, and we have revised lines 50 and 59 in the re-uploaded revised manuscript.

Comments 6. qRT-PCR should be RT-qPCR.

Response 6: Thank you for your suggestion and we agree with your comment. We have changed qRT-PCR to RT-qPCR where it appears in the manuscript and marked it with red font.

Reviewer 2 Report

Comments and Suggestions for Authors

Review on ijms-3049760

The study is interesting and provides well-supported conclusions based on a wide range of up-to-date methodologies. However, there are deficiencies and many mistakes both in the description and statistical analysis of the experimental data. Format mistakes are also widespread. Therefore, it can only be accepted for publication after MAJOR REVISION.

Citations are not proper, e.g. in Line 34, instead of “cold and frost damage12.”, it should be written as “cold and frost damage [1,2].” It must be corrected throughout the MS. In Lines 50 and 59, “Error! Reference source not found.” must be deleted.

L77: the genome should be given

L85: A gene cannot be better adapted to a condition, a plant carrying a specific allele might be, but not a gene

L88: “genetic breeding” seems to be a tautology, I guess breeding is enough here

L91: HMM should be explained here and not in Line 411

L98: Soloist is a subfamily of AP2/ERF family 40 of transcription factors, so its mentioning in the present form does not make sense in this sentence, it must be completely rephrased

L101: Soloist should be capitalized

L108: names of genes must be italicized

L108: Motif should not be capitalized

L121: “The structural composition of the genes showed that 51.16% of the genes contained exons, while the remaining genes contained at least one intron.” this sentence is absurd: you do not mean that 49% of the genes do not contain exons. It must be corrected, “contained only exons” or “did not contain intron”?

L202: freezing temperatures are also cold, here “chilling” should be used for 4C

L211: “22 significantly expressed genes selected by qRT-PCR” should be modified as “22 significantly expressed genes to check their expressions by qRT-PCR

L213: the authors should be clear: cold is not useful, freezing or chilling, what do you mean?

L223: this is chilling, not cold

L244: Student’s t-test is inappropriate here please use ANOVA and post-hoc tests to show all significant differences. No such differences are shown on the graphs, which is not likely to be correct. The authors must repeat the whole statistical evaluation of the results using adequate tests

L294: citation is missing here

L324: A statement without any support. Citation?

L389: “Proline is a non-protein amino acid widely found in plants” Are you sure of that??? Proline is a crucial AA in proteins!!!

L398: A gene cannot be resistant to stress, a plant carrying a specific gene/allele might be, but not a gene

L404: a period is missing from the end of the sentence

L405: the same sentence is in double copy

L456: the basic parameters of RNAseq analysis must be given here

L484: how actin was proved to be a good internal control gene for the Real-Time PCR analyses?

Comments on the Quality of English Language

Minor editing will be needed, please check my detailed evaluation.

Author Response

Dear reviewers

Thank you very much for your comments on the manuscript. Based on your suggestions, we have tried our best to revise the relevant parts and made some changes to the manuscript. These changes will not affect the content and framework of the paper. We have answered all the questions you raised below. Here, we have listed the changes and highlighted them in red in the revised paper.

Comments: Citations are not proper, e.g. in Line 34, instead of “cold and frost damage12.”, it should be written as “cold and frost damage [1,2].” It must be corrected throughout the MS. In Lines 50 and 59, “Error! Reference source not found.” must be deleted.

Response : Thank you very much for your suggestion, we have revised this issue in the revised manuscript

Comments: L77: the genome should be given

Response : Thank you for your suggestion and we accept this comment. The genomes used in this paper have been provided within the Supplementary file

Comments: L85: A gene cannot be better adapted to a condition, a plant carrying a specific allele might be, but not a gene

Response : Thank you very much for your suggestion, we have reworded the sentence in line 87

Comments: L88: “genetic breeding” seems to be a tautology, I guess breeding is enough here

Response : Thank you very much for your suggestion, we have reworded the sentence in line 91

Comments: L91: HMM should be explained here and not in Line 411

Response : Thank you for your suggestion, we have made a change to the problem that appeared in line 95

Comments: L98: Soloist is a subfamily of AP2/ERF family 40 of transcription factors, so its mentioning in the present form does not make sense in this sentence, it must be completely rephrased

Response : Thank you for your suggestion, we have corrected the words that appear in line 101

Comments: L101: Soloist should be capitalized

Response : Thank you for your suggestion, we have corrected the words that appear in line 105

Comments: L108: names of genes must be italicized

Response : Thank you for pointing this out, , we have corrected this error in Figure S1.

Comments: L108: Motif should not be capitalized

Response : Thank you for your suggestion and we agree with your comment. We have corrected and flagged the error in line 113 and onwards.

Comments: L121: “The structural composition of the genes showed that 51.16% of the genes contained exons, while the remaining genes contained at least one intron.” this sentence is absurd: you do not mean that 49% of the genes do not contain exons. It must be corrected, “contained only exons” or “did not contain intron”?

Response : Thank you very much for pointing this out, we have rephrased the sentence that appears in line 126

Comments: L202: freezing temperatures are also cold, here “chilling” should be used for 4C

Response : Thank you very much for pointing this out, we have rephrased the sentence that appears in line 208

Comments: L211: “22 significantly expressed genes selected by qRT-PCR” should be modified as “22 significantly expressed genes to check their expressions by qRT-PCR”

Response : Thank you very much for your suggestion, which we accept and have revised in line 217 of the manuscript.

Comments: L213: the authors should be clear: cold is not useful, freezing or chilling, what do you mean?

Response : Thank you very much for your suggestion, which we accept and have revised in line 219 of the manuscript.

Comments: L223: this is chilling, not cold

Response : Thank you very much for your suggestion, which we accept and have revised in line 224 of the manuscript.

Comments: L244: Student’s t-test is inappropriate here please use ANOVA and post-hoc tests to show all significant differences. No such differences are shown on the graphs, which is not likely to be correct. The authors must repeat the whole statistical evaluation of the results using adequate tests

Response : Thank you for your suggestion and we have modified Figure 7.

Comments: L294: citation is missing here

Response : Thank you very much for your suggestion and we have added a new reference to what appears on line 306.

Comments: L324: A statement without any support. Citation?

Response : Thank you very much for your suggestion and we have added a new reference to what appears on line 329.

Comments: L389: “Proline is a non-protein amino acid widely found in plants” Are you sure of that??? Proline is a crucial AA in proteins!!!

Response : Thank you very much for pointing this out, we have rephrased the sentence that appears in line 405.

Comments: L398: A gene cannot be resistant to stress, a plant carrying a specific gene/allele might be, but not a gene

Response : Thank you very much for pointing this out, we have rephrased the sentence that appears in line 414.

Comments: L404: a period is missing from the end of the sentence

Response : Thank you very much for pointing this out, we have rephrased the sentence that appears in line 420.

Comments: L405: the same sentence is in double copy

Response : Thank you very much for pointing this out, we have rephrased the sentence that appears in line 422.

Comments: L456: the basic parameters of RNAseq analysis must be given here

Response : Your suggestion is sincerely appreciated and we have revised line 471 of the manuscript to refine this point.

Comments: L484: how actin was proved to be a good internal control gene for the Real-Time PCR analyses?

Response : Thank you for pointing this out, and we have added a reference to this in line 502 of the manuscript.

Round 2

Reviewer 2 Report

Comments and Suggestions for Authors

It can now be accepted for publication.

Author Response

Comments: AS the reviewer 1 pointed out, it is essential to confirm the expression of FvERF23 in the transgenic Arabidopsis plants. The authors responded to this point by stating that the identification of the expression levels of the FvERF23 gene we provide in the Non-published Material of the corrected manuscript. However, no data were provided, and the results were never mentioned in the main text.

Response: 

Dear Editor, As requested, we have resubmitted the revised manuscript incorporating new content highlighted in green within the main text, along with an additional figure included in Supplementary File Figure S3.